# The SKMT Algorithm: A method for assessing and comparing underlying protein entanglement

**Arron Bale** [1]*, **Robert Rambo** [2], **Christopher Prior** [1]

**1** Department of Mathematical Sciences, Durham University, Durham, United Kingdom, **2** Diamond Light Source, Harwell Science and Innovation Campus, Didcot, United Kingdom

* arron.n.bale@durham.ac.uk

## Abstract

We present fast and simple-to-implement measures of the entanglement of protein tertiary structures which are appropriate for highly flexible structure comparison. These are performed using the SKMT algorithm, a novel method of smoothing the $C^\alpha$ backbone to achieve a minimal complexity curve representation of the manner in which the protein's secondary structure elements fold to form its tertiary structure. Its subsequent complexity is characterised using measures based on the writhe and crossing number quantities heavily utilised in DNA topology studies, and which have shown promising results when applied to proteins recently. The SKMT smoothing is used to derive empirical bounds on a protein's entanglement relative to its number of secondary structure elements. We show that large scale helical geometries dominantly account for the maximum growth in entanglement of protein monomers, and further that this large scale helical geometry is present in a large array of proteins, consistent across a number of different protein structure types and sequences. We also show how these bounds can be used to constrain the search space of protein structure prediction from small angle x-ray scattering experiments, a method highly suited to determining the likely structure of proteins in solution where crystal structure or machine learning based predictions often fail to match experimental data. Finally we develop a structural comparison metric based on the SKMT smoothing which is used in one specific case to demonstrate significant structural similarity between Rossmann fold and TIM Barrel proteins, a link which is potentially significant as attempts to engineer the latter have in the past produced the former. We provide the SWRITHE interactive python notebook to calculate these metrics.

## Author summary

There is much interest in the development of quantitative methods to compare different protein structures or identify common substructures across protein families. As our understanding of the flexible and dynamic nature of protein structures advances it will be necessary to develop methods for comparing protein structure which accounts for this

**Data Availability Statement:** Python notebook available via a github repository at https://github.com/arronelab/SWRITHE.

**Funding:** This work was completed as part of a joint EPSRC MoSMed CDT (EP/S022791/1) and

Diamond Light Source Ltd. studentship award to AB. The funders had no role in study design, data collection and analysis, decision to publish, or preparation of the manuscript.

**Competing interests:** The authors have declared that no competing interests exist.

flexibility. This can be achieved by assessing and comparing the underlying shape of protein structures which are not obfuscated by the small scale (primary and secondary) complexity of the structure, and instead focus on their large scale (tertiary) entanglement. Here we present such a novel set of quantitative measures by smoothing and simplifying the amino-acid backbone into a minimal representation of its true flexibility. We demonstrate these measures of a protein chain's self-entanglement have a number of critical properties which make them potentially impactful. First, by studying the distribution of entanglement across a wide sample of proteins, we show that there exists a minimum expected amount (a lower bound) of entanglement given the protein's length. This bound is shown to be useful in ensuring realistic predictions from experimental structural determination methods. Second, using fundamental properties of this entanglement measure, we identify the presence of helical structures across various length scales in proteins, which provide stability to the structure. Third, we show they can be used to highlight significant structural similarity between two families of proteins currently classed as distinct, but which have been shown to share a surprising experimental link. Finally, we provide an interactive python notebook to compute these measures for a given protein.

## Introduction

The sequence of amino acids which form a protein is its *primary* structure and it is always identifiable. Researchers often visualise a protein's global structure via its $C^\alpha$ backbone curve: the discrete 3-dimensional curve whose points represent the central $\alpha$-carbon atom of each amino acid residue. Secondary structure is the term given to the shape of local segments of the protein's backbone curve. The two most clearly defined types of secondary structure are $\alpha$-helices and $\beta$-strands, both of which are helical and relatively uniform across proteins [1]. Other secondary structures we collectively refer to as linkers, which can have a much higher degree of variation across proteins. The variation of these linker sections allows the protein to form complex global entanglements, referred to as the *tertiary* structure, which determine the functionality of the protein [2].

The largest known experimentally determined tertiary structure database is the Protein Data Bank (PDB [3]) which currently contains over 200,000 entries. There has been much work aimed towards classification of the tertiary structures within this database, for example the CATH [4], SCOP [5], and Dali [6] classifications. The families identified in these classifications give greater insight into the correlation between tertiary structure and function. One such example is the TIM-barrel, which is a conserved fold shown to have a consistent binding site across its family [7].

The advent of machine learning (ML) methods such as AlphaFold [8] and RosettaFold [9] has opened the possibility of routinely predicting tertiary structure, and expanding the collective database beyond the existing experimentally determined structures. The progress of these methods is staggering, in the most recent data release AlphaFold2 was used to predict the structure of over 200 million amino acid sequences deposited in UNIPROT. ML methods are trained to identify the relationship between primary sequence and tertiary structure, using the experimentally determined structures available in the PDB. However, when in their natural environment (rather than in crystal structure formation) proteins often exhibit significant flexibility, adopt multiple differing configurations or, in the extreme, display intrinsic disorder. As a result, ML methods based on a static picture of protein structure can struggle with proteins that are far more flexible and dynamic in nature. In addition, where a sequence is not well

characterised in the existing data bank the ML prediction will be significantly uncertain, and in many cases, meaningless. With this in mind the aim of this study is to provide metrics to compare and characterise tertiary structures in a manner which would be less affected by small or even medium scale motion, or the exact geometry of the local (secondary) structure elements of the protein.

An example of the similarity we are looking to characterise is shown in Fig 1A and 1B. The trefoil curve Fig 1A is knotted in the sense that it must be cut in order to deform it into a circle. The curve shown in Fig 1B can obtained by locally distorting Fig 1A continuously without the curve crossing itself, i.e without any such cutting. Thus in some sense they are folded in a similar fashion, but this would not be captured by the standard distance based metrics used for rigid structure comparison. A more pertinent example with real protein structures is shown in Fig 1C and 1D where the smoothed $C^\alpha$ backbones of two proteins which have differing CATH classifications, a TIM barrel Fig 1C and a Rossmann fold Fig 1D, can be seen to bear a striking similarity up to such a distortion. They have clear helically coiled domains with the same number of coils in each domain and similar relative orientations of these domains. The curves shown are smoothed using the SMKT algorithm which we introduce in this study, but for now it suffices to say this smoothing removes the local secondary structure geometry, and in doing so reveals the similarity of these global folds.

The specific example in Fig 1C and 1D is more than just a curiosity. The results of [10] indicate a very close link between these two domains on the primary sequence, or even evolutionary level. In [10] the authors used directed evolution techniques in attempts to design a TIM barrel structure, but found instead a Rossmann fold-like structure was produced. When testing these results against the contemporary state of the art computational techniques, none were able to predict the produced structure and most agreed that the sequence should indeed produce a TIM-barrel conformation. Both TIM barrels and Rossmann folds have a $\beta$ sandwich structure (anti-parallel strand helices), whose cross-bonds provide stability to the structure. The helical sections of the two figures result from this $\beta$-sandwich motif. In this study we aim

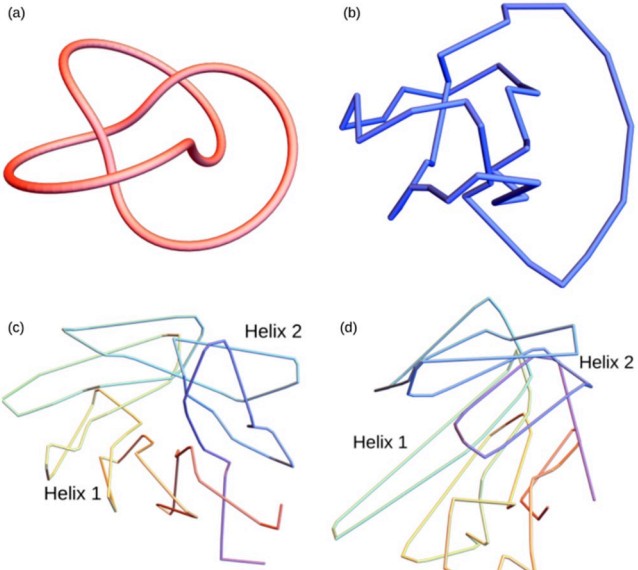

**Fig 1. Depictions of the notion of structural similarity we seek to quantify in this study.** In both examples the topological similarity would be missed by distanced based metrics.

**Table 1. Examples of distance alignment based metrics applied to the two proteins in Fig 1.** For reference an RMSD $\leq 2$Å and TM-score $> 0.5$ would be considered significantly similar.

| Method | RMSD (Å) | TM-score |
|---|---|---|
| DALI [6] | 5.5 | - |
| TM-align [11] | 5.14 | 0.35 |
| jFATCAT [12] | 8.48 | 0.25 |

to show this large scale helical conformation is highly prevalent in a wide variety of protein structures, often of a similar scale across a variety of CATH families. We believe identifying similar size helical structures could provide insight into de-novo design methods.

The most commonly used methods of quantitative structural comparison are based on root mean squared comparisons [13]. Two structures are rotated such that the sum of the minimum Euclidean distance between sequentially aligned points is minimised, the so-called root-mean-squared distance (RMSD). Very often the comparison is between similar subsections, for example one can compare hinging proteins this way. However, alignment based distance metrics, despite their many merits, are not suited to the types of similarity we are aiming to quantify in this study. To highlight this, we apply some of the readily available alignment methods to the proteins in Fig 1C and 1D. The results in Table 1 show that these metrics miss the similarity that is visually apparent. The DALI method [6] is an alignment based method which seeks the largest common substructure between the two proteins. TM-align [11] uses an RMSD based metric which is less sensitive to local variations than pure RMSD, called TM-score [14]. The jFATCAT method [12] uses a more flexible approach to alignment, allowing a certain number of twists in the backbone to align residues. That none of these methods find a strong similarity between our example proteins is not a surprise. The relative alignment of their $\beta$-strands differs greatly, leading to their distinct CATH classification. However, the similarity of their helical subsections, including the number of helical turns, is clear to see.

Another class of shape classification metrics, topological metrics, are derived from aspects of Knot theory [15, 16]. The main advantage of such metrics is that they are invariant to rotations and translations and therefore do not require alignment. A second advantage that is crucial to the aim of this study is that they classify structures up to isotopy, *i.e.* they measure two structures as similar if they can be distorted into each other without having to construct the distortion. This is exactly the notion of flexible similarity outlined above. Many topological metrics proposed in the literature have been designed to detect knotting in proteins, e.g. [17–19]. However, this approach loses some information about the specific arrangement of secondary structures, which is vital for our purposes. For example, they are not designed to detect the similar helical structure seen in Fig 1C and 1D. The aim of this study is to develop measures that are topological in nature, and able to easily detect helical structures. So we turn to a related set of quantities based on the writhe, a quantity which arose from the study of DNA-like ribbon topology and which is commonly used as a measure of DNA supercoiling [20–23].

## The writhe and its use characterising proteins

The formal mathematical definition of the writhe of a three-dimensional curve $\mathbf{x}$ with tangent vector $\mathbf{T}$ is given by the Gauss linking integral [24]

$$Wr = \frac{1}{4\pi} \int_{\mathbf{x}} \int_{\mathbf{x}} \mathbf{T}(s) \times \mathbf{T}(t) \cdot \frac{\mathbf{x}(s) - \mathbf{x}(t)}{\|\mathbf{x}(s) - \mathbf{x}(t)\|^3} \, \mathrm{d}s \, \mathrm{d}t. \tag{1}$$

**Fig 2. Intuitive interpretation of the writhe; an average of the sum of signed crossings over all projections.**

In the protein context **x** would represent the protein's $C^\alpha$ backbone. As proteins are discrete curves whose points represent the $C^\alpha$ atoms, we use the discrete analogue of this double integral given in [25].

For an intuitive understanding of the writhe calculation, consider the oriented trefoil curve seen in Fig 2A. Viewed from this direction, one can count four negative crossings, with sign given according to Fig 2C and 2D. The signed sum of these crossings is therefore −4, representing an oriented measure of the amount the curve wraps around itself. In Fig 2B we see the **same** curve from a different direction, with now only three crossings visible, again with negative sign. It can be shown [26] that Eq 1 is the average of the number of crossings taken over all possible directions (it is approximately 3.52 for this example).

The first application of the writhe to protein structures of which we are aware is in [27], where it is used to confirm the correct threading of the N-terminus of the bovine pancreatic trypsin inhibitor protein in simulation. This study utilised the key property that the writhe jumps by multiples of 2 when a curve section crosses through itself to detect valid pathways. This makes it an excellent metric for detecting significant changes in knotting/entanglement which would be missed by RMSD based metrics (as clearly argued in [28]). Writhe-based metrics have been developed further in [28, 29], where generalisations of the writhe integral are used to effectively classify proteins in agreement with the CATH database to a high degree of accuracy. The generalisations are the so called higher order Gauss integrals. If the integrand density of Eq 1 is written as $w(s, t)$, then the higher order variants are of the form:

$$\int_{\mathbf{x}}\int_{\mathbf{x}}\cdots\int_{\mathbf{x}} w(s_1, t_1) w(s_2, t_2) \ldots w(s_n, t_n) \mathrm{d}s_1 \mathrm{d}t_1 \mathrm{d}s_2 \mathrm{d}t_2 \ldots \mathrm{d}s_n \mathrm{d}t_n. \tag{2}$$

Unlike the standard writhe these quantities can be somewhat difficult to interpret. Intuitively it can be thought of as the number of times the directions of labelled pairs of crossings, say $(s_1, t_1)$ $(s_2, t_2)$, coincide (see [29] for a more detailed discussion). The authors use up to 30 of these integrals to create a distance metric for comparing structures. In [30] a comparison metric based on the writhe value of subsets of the proteins of fixed lengths was combined with a number of other quantities to create a comparison metric against the SCOP classification, again with good results. In [31] a more rapid metric was created from the same Gaussian integrals as in the other two studies, once again shown to compare favourably to the SCOP benchmark set. The measures introduced in [29] were recently developed further in [32] where the Gaussian-integral approach was applied to subsections (fragments) of the protein, via a fingerprinting technique, using the entanglement of sub-chains to identify rare conformations in proteins. We highlight two aspects of these works that motivate developing our own approach. First, despite their evident success, there is an element of difficulty in interpreting these higher order writhe integrals intuitively. Second, we want to identify structural similarities which would be missed by standard classification methods, such as the Rossmann fold/TIM-barrel example discussed above. Our approach is instead to apply an appropriate smoothing (SKMT

method) such that the simpler first order (standard) writhe measure can yield critical information about a protein's tertiary fold.

The final aspect of the writhe literature we build upon in this is study is whether there are limits on the amount of entanglement of protein backbones as measured by the writhe. A theoretical upper bound on the writhe of smooth thick knots is presented in [33]:

$$|Wr(K)| \leq \frac{1}{4}\left(\frac{L}{R}\right)^{4/3},\tag{3}$$

where $L$ is the knot $K$'s length and $R$ its radial thickness. The basic idea being that longer knotted curves (relative to their width) can achieve more complex configurations and hence higher entanglement (writhe), which is why the bound grows super-linearly. Given proteins have distinct end points (*i.e* are not mathematically knots) and are discrete (*i.e* not smooth) curves, this bound will not strictly apply to such curves. In addition the notion of radius is difficult to define for proteins (although some attempts have been made in the past [34]). That being said we will show this bound has some clear relationship to protein structures, but that for sufficiently large proteins, the extra length does not produce more complex configurations and instead the writhe tends to grow linearly. In related work the relationship between the writhe and length of uniform random walks was studied in [35], and then applied to proteins in [36]. However, since the local geometry of the protein's backbone is tightly constrained by the Ramachandran steric constraints [1], we may expect that a random walk cannot capture the systematic entanglement of a protein's backbone, a supposition we investigate in this study.

The novelty of our work in this field relates to a very important characteristic of proteins: secondary structure elements are highly rigid on the global scale, which significantly constrains the potential for creating complex global entanglement [37]. The SKMT backbone algorithm we develop explicitly aims to capture the relative rigidity of different sections of the protein through simplified curves. We then show writhe measures applied to these curves better characterise the relationship between the protein's secondary structure and its permissible entanglement than other possible choices.

## Aims

To summarise, in this study we present a method for smoothing and simplifying the amino-acid backbone into a minimal representation of its true flexibility (the SKMT method). We then derive writhe based metrics for this smoothed backbone curve which can be used for the following aims:

1. To class structures as similar if they can be distorted into each other without significantly changing the topology of the fold, *e.g.* preserving knotting type or large scale helical structure.

2. To pick up potentially meaningful structural similarities missed by the standard classifications.

3. To provide clear bounds on the amount of entanglement in protein structures relative to their secondary structure.

As discussed above, the first aim has been addressed in previous studies [28–31], however, aims 2 and 3 provide constraints to the metrics not met in previous studies. The third aim was partly developed with a concrete application in mind: to ensure tertiary structure searches for relatively low-resolution experimental techniques such as small angle biological x-ray scattering can be restricted to plausible structures.

In the methods section we discuss the two main quantities we calculate, the writhe and absolute crossing number. We also discuss the motivation and method for our smoothing of the $C^\alpha$ backbone whilst preserving its fundamental topology: the SKMT method. In the results section we first show both quantities have highly restrictive empirical bounds across a large ($> 10000$) sample of structures which differ significantly at the primary sequence level. We then show how helical super-structures and knotting are the main ways of developing systematic complexity in structures. We demonstrate there is a very common super-helical scale in tertiary structures shared across a variety of CATH domains, and that the introductory example similarity of the TIM barrel and Rossmann fold is indicative of a consistent similarity between these two fold types. We then show how the unsigned writhe bound can be used to restrict expansive structural search methods for BioSAXS data. Alongside this study we provide the SWRITHE package in the form of an ipython notebook which can be found at (https://github.com/arronelab/SWRITHE).

## Methods

First we introduce the basic quantities calculated by the SWRITHE notebook.

### The writhe

We consider a discrete curve $\mathcal{C}$ of length $j$, characterised by a set of three dimensional coordinates $\mathbf{x}_k$ i.e. $\mathcal{C} = \{\mathbf{x}_k\}_{k=1}^{j}$. We calculate the writhe of a subset of the curve $\mathcal{C}_{in} = \{\mathbf{x}_k\}_{k=i}^{n}$ through the following formula:

$$Wr(\mathcal{C}_{in}) = 2\sum_{l=i+1}^{n} \sum_{m=l+1}^{n} \frac{\Omega_{lm}}{4\pi}, \tag{4}$$

where $\Omega_{lm}$ is a signed spherical area representing the contribution to Eq 1 from the crossing of edges connecting the pairs of points $(l-1, l)$ and $(m, m-1)$. There are various formulae for $\Omega$, we use method (1a) of [38]. We calculate spherical areas since crossings can be represented as points on the unit sphere and the writhe represents a signed area covered on this sphere (it can include multiple full coverings of the sphere), so the areas represent signed crossing densities [38]. The function `calculate_writheFP` calculates $Wr(\mathcal{C}_{in})$ for all $1 \leq i \leq j - 5$, $i + 5 \leq n \leq j$ for a given curve $\mathcal{C}$ (we need at least 5 points for a meaningful writhe calculation).

### The average crossing number

For bounds on entanglement in particular it is useful to count the number of crossings without sign as a positive definite measure of complexity of the fold, we call this the average crossing number (*acn*):

$$acn(\mathcal{C}_{in}) = 2\sum_{l=i+1}^{n} \sum_{m=l+1}^{n} \frac{|\Omega_{lm}|}{4\pi}, \tag{5}$$

The bound in [35] relates to this quantity. The function `calculate_writheFP` also calculates $acn(\mathcal{C}_{in})$ for all $1 \leq i \leq j - 5$, $i + 5 \leq n \leq j$ for a given curve $\mathcal{C}$.

### Smoothing by secondary structure: The SKMT algorithm

**The rationale.**   We are aiming for a measure of the entanglement of protein backbones which does not include the secondary structure's inherent helical nature. This aim can be achieved by smoothing or simplifying the backbone curve representation. One suggestion

from the literature has been to sample the full $C^\alpha$ backbone curve of the protein evenly every $n$ amino acids, as in [31]. A second suggestion has been to minimise the curvature of the smoothed backbone, as in [39]. However, we also seek a measure which has a sensible growth in complexity with respect to length of the representative curve. We argue the way to achieve this second aim is to represent each secondary structure element (SSE) as a minimal number of edges of a discrete curve.

Indeed, we aim to demonstrate that it is not the number of amino acids composing the molecule which determine the potential complexity of the structure's fold, rather the number of distinct SSEs it has, and in particular the number of linker sections. In the case of $\alpha$-helices and $\beta$-strands, when their locally helical structure is ignored their contribution to the overall tertiary fold is essentially that of a single inflexible edge. It is the more structurally variable linker sections which allow the structure to fold, so the greater the number of these sections there are, the higher the potential global complexity. In the SKMT algorithm we represent the helical secondary structures with a single edge. We would like to argue the same for the more structurally varied linker sections. Their main functionality is to link the other SSEs (as in $\beta$ sheets) and often the specific geometry of the linker is less important than the orientation of its end points. However, there are occasionally linkers which coil around other SSEs leading to knotting (or slip knotting and the other various rare but complex entanglements found in some proteins [40]). Replacing them by a straight edge will miss this essential entanglement. In this case, we adapt the KMT algorithm [41] to reduce the linker to its minimal representation which preserves any such knotting, as illustrated in Fig 3.

**The SKMT method.**   We assume as input the coordinates of the $C^\alpha$ backbone from the PDB file and a secondary structure assignment, usually from PSIPRED [42]. We then construct a new discrete curve from the full $C^\alpha$ backbone as follows.

1. Take the $C^\alpha$ coordinates of the N-terminal amino acid.

2. For each SSE, we apply the KMT algorithm [43] locally to this section. That is, if the triangle defined by three sequential points of the SSE is not intersected by any edge of the rest of the curve, then we can safely remove the middle point of this triangle. See Fig 3 for an example of such an intersection.

3. Repeating this for each SSE, we reduce the backbone to a minimal representation of its SSEs which preserves any essential entanglement.

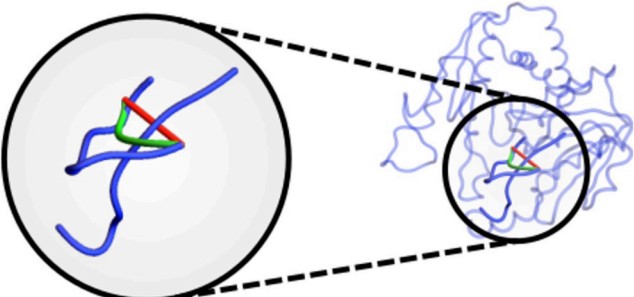

**Fig 3. An example of the non local coiling that is preserved via the SKMT algorithm.** In blue we see the backbone of the trefoil knotted acetylornithine transcarbamylase (PDB 3KZK). In red we highlight the straight edge connecting consecutive SSEs, which passes under the C terminus. In green, we see the edge connecting these SSEs output by the SKMT algorithm, which preserves the non local entanglement, and therefore the knottedness of this curve.

We call this the SKMT (Secondary KMT) algorithm in what follows. It differs from the original KMT algorithm which acts on the whole curve, trying to simplify it to as few points as possible (for example to calculate computationally expensive knot invariants [43]). By contrast the initial separation of the curve into secondary structures to which the KMT algorithm is applied distinctly forbids such a dramatic simplification, and will preserve non-knotting complexity such as large-scale super-helical coiling. The SKMT algorithm can be applied to a selected PDB file using the `skmt` function.

**SKMT length.**   We shall commonly calculate distributions $Wr(\mathcal{C}_{1n})$ and $acn(\mathcal{C}_{1n})$ of the SKMT smoothed curves $\mathcal{C}$ as a function of $n$, the number of points of the subsection $(1, n)$. This implicitly associates a "length" of the protein with the number of points of its SKMT curve. The variability of the section sizes in the SKMT smoothed curve means this length measure will **not** be the arclength of the curve. We will argue later this choice is a good measure of length with regards to the aim of finding clear and consistent empirical bounds on the values of both $Wr$ and $acn$ of protein backbones as a function of their secondary structure. With this in mind we define the length of an SKMT smoothed backbone curve to be simply the number of points of the smoothed curve and will aim to justify this choice.

**An illustrative example.**   To illustrate the reasoning behind our choice of backbone smoothing and definition length, we consider two relatively large proteins, both consisting of over 550 residues but with significantly different numbers of distinct secondary structures, shown in Fig 4. The protein in Fig 4A, is made up of 567 amino acids and 80 relatively short secondary structure elements (both $\beta$-strands and $\alpha$-helices). The protein in Fig 4B has more amino acids (583), but significantly fewer secondary structure elements (57), as it comprises some long $\alpha$-helical sections. As shown in Fig 4C the value $acn$ for the full curve of the first structure is significantly larger than that of the second structure and its SKMT length also significantly larger. In particular, the full curve $acn$ value of the first structure is around 1.65 times greater than that of the second structure, which is proportional to the ratio of the number of points of their SKMT smoothed backbones (104 and 68 respectively). As noted though, the first structure is actually slightly shorter in terms of its number of amino acids. If we were instead to have smoothed their backbones by sampling every $n$ amino acids (for $n$ sufficiently small), these long $\alpha$-helical sections would be sampled multiple times and the relationship

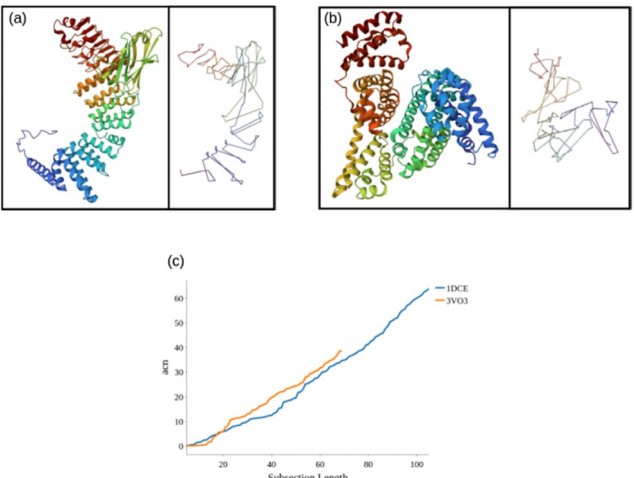

**Fig 4. An *acn* comparison of two proteins which have similar primary sequence length, but significantly different number of secondary structure sections and therefore possible complexity.**

between complexity and length of the smoothed curve would not be so clear. However, by smoothing via the SKMT algorithm, we represent the global entanglement's relationship to the relative flexibility of the protein due to its secondary structure. This is one illustrative example and not the sum total of our argument. We provide further justification of this scaling relationship being meaningful in what follows.

### Helical structures and linear writhe growth

One key property of the writhe that is useful to our study is its link to helical geometries. Helices have a uniform writhe density due to their consistent chiral-coiling, so $Wr(\mathcal{C}_{1n})$ plotted as a function of subsection length $n$ is a straight line whose gradient can be given in terms of the helix's pitch $P$ and radius $R$. The writhe per turn is given in [44]:

$$Wr = 1 - \frac{P}{\sqrt{P^2 + 4\pi^2 R^2}} \tag{6}$$

In Fig 5A we see an example of an SKMT smoothed $C^\alpha$ backbone, which has a globally helical geometry. In Fig 5B we see a plot of $Wr(\mathcal{C}_{1n})$, the writhe of the smoothed curve as a function of its length. There is a clear almost linear rise due to the helical nature of the structure. Indeed, this structure has radius 7.58 and pitch 5.98 (found by approximating the helical axis), giving a writhe per turn of 0.88. Considering the substructure relating to the linear portion of the graph (that is between $n = 30$ and $n = 130$) there is a rise in writhe of 12. We compare this to the per turn result by inspecting the structure, for which we count 14 clear turns. Since $14*0.88 = 12.3$, it is clear this writhe calculation is accurately quantifying the consistent helical superstructure of the protein. We shall see later that this geometry is common to sub units of a large number of proteins, spanning many CATH domains.

## Results

### Length constraints on the writhe of proteins

To study the distribution of writhe amongst proteins, we take a representative sample of proteins from the PDB with the following criteria:

1. A good resolution (<2Å)

2. Consisting of between 30–300 residues.

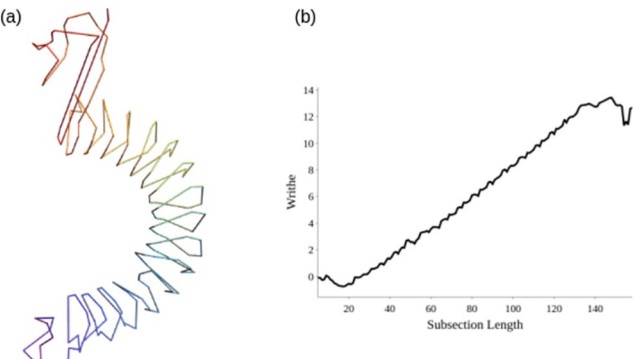

**Fig 5. An example of a protein that sees linear growth in writhe due to its globally helical structure.**

3. Redundancies removed at 70% sequence identity

4. Good model quality: $R_{work} \in [0, 0.2]$ and $R_{free} \in [0, 0.25]$.

This yielded 10736 entries from the PDB. We take the first monomer unit from each PDB file, and computed its SKMT smoothed backbone. The values of $Wr(\mathcal{C})$, the writhe of the whole SKMT curve, are presented in Fig 6 as a function of the SKMT length $L$. There are a number of critical aspects of this plot we highlight:

1. 99.9% of all values lie within the curves

$$\pm \frac{1}{4} \left( \frac{L}{R} \right)^{4/3},\qquad(7)$$

the theoretical knot bound from [33], with a radius $R = 2.7$ (coinciding with the mean "tube" thickness of 2.7Å found in [45] for minimal $C^\alpha$ triplet radii).

2. For larger proteins, $L > 35$, the $Wr$ values increasingly fail to get close to this limit. We find 97.9% of the structures fit within a linear bound $0.12L$.

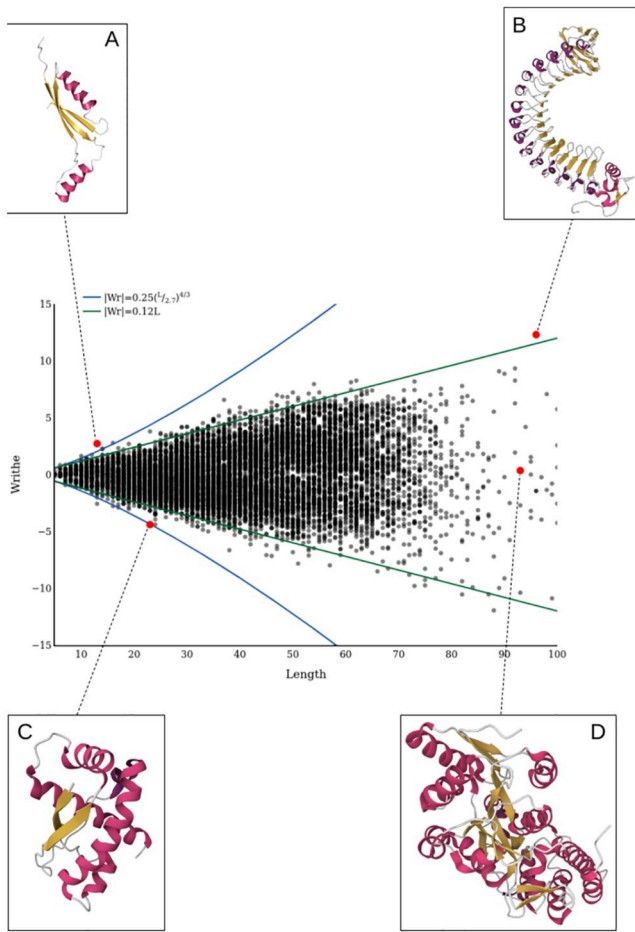

**Fig 6. The distribution of writhe for a representative sample of $>$10000 proteins from the PDB in black.** In blue we see the theoretical writhe bound [33] with $R = 2.7$. In green, we see linear growth in writhe with a gradient of 0.12. Inset: A: PDB entry 1VQ3. B: PDB entry 2OMZ. C: PDB entry 2RH3. D: PDB entry 4O4B.

3. A significant number of proteins have a low or even close to zero writhe, consistent across all scales.

We now highlight some exemplar proteins from this distribution. In Fig 6A we see a small protein which is one of the few whose writhe exceeds the theoretical knot bound, we will discuss this shortly after highlighting some other aspects of the distribution. The knot "bound" assumes that the maximum complexity available to a curve as measured by the writhe increases with length, whilst a linear trend potentially indicates a fixed maximum complexity independent of length due to helical geometries. Indeed the helical protein shown in Fig 5, which is the same structure as shown in Fig 6B, can be seen to be very close to this linear bound. The $\beta$-strands of the protein seen in Fig 6B are all parallel, separated by a linker-$\alpha$ helix-linker pattern. As a result, there is a consistent handedness of winding along the length of the protein, leading to the linear build up of writhe. According to the CATH topology classification [4] this protein is an example of a Leucine Rich Repeat (LRR) Right-handed Beta-Alpha superhelix. In particular, it is a tandem repeat domain, a family of proteins that are of keen interest due to their protein-protein interactions [46].

Fig 6C shows a relatively small protein whose writhe does exhibit super-linear growth, falling right on the knot bound. This protein builds up significant writhe due to its trefoil knotted structure as per [47]. By contrast Fig 6D highlights that it is difficult for larger proteins to systematically build up writhe. Visually it appears to have a complex entanglement, but the total writhe of this protein is close to 0. In Fig 7A we see that $Wr(\mathcal{C}_{1n})$ has a peak value around 1.5 corresponding to the locally helically coiled initial substructure highlighted in Fig 7B. In Fig 7C we see two counter helical loops which contribute no net writhe due to the cancellation in the signed sum. Finally, the long subsection passing through the rest of the structure highlighted in Fig 7D leads to a cancellation of the accumulated writhe. The patterns of complexity seen in these substructures is common across the data set, and the accompanying SWRITHE notebook can be used to identify both helical and net zero writhe substructures in this fashion.

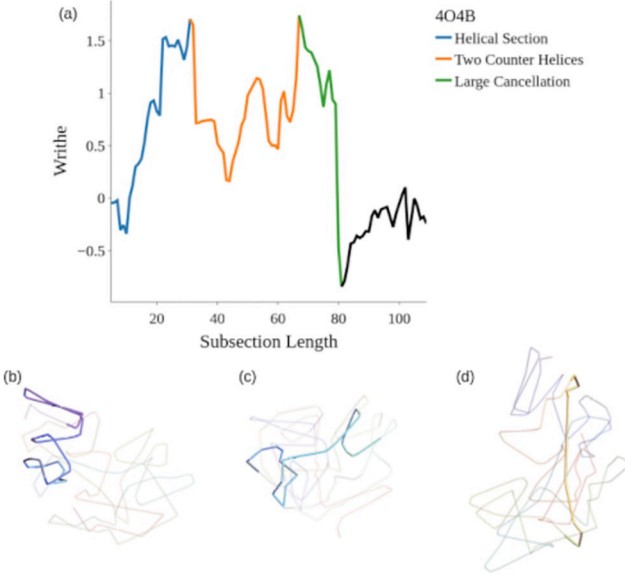

**Fig 7. An example of the potential domains present in a complex entangled yet net zero writhe structure.**

We now discuss some additional tests we performed to verify these results are meaningful.

**N-termini and tags.**   It is often the case that the N-terminus may be missing from a PDB entry, or a purification tag is used during the experimental determination of the structure. To check the possibility such occurrences are not affecting our results, we re-ran this numerical experiment randomly deleting some of the amino acids from the N-terminus end of the original PDB entry. In S1 Fig we find this does not affect our results.

**Is this definition of length a good one.**   As discussed above, the length $L$ on the $x$-axis is the number of points of the SKMT smoothed curve. There are a number of other options we could have chosen, including but not limited to:

1.  The arclength of the SKMT curve.

2.  The number of amino acids of the protein.

3.  The length of another appropriately smoothed curve (say sampling every $n$ atoms).

We consider the distribution of writhe against length for each of these possibilities in S2, S3 and S4 Figs. It is shown that none of these other choices lead to a consistent empirical "bound" on the value of $Wr$ as a function of the alternative length definition. In these cases it tends to be that a linear bound that works for shorter length curves does not tightly constrain (most of) the larger curves, or vice-versa. It is only with the SKMT curve length (number of points) that we find we can draw linear bounding curves which tightly contain the vast majority of the structures across all lengths. We cite this as justification for our choice of SKMT smoothing as correctly capturing the growth in complexity of the protein's tertiary structure with a minimal representative curve.

## A brief discussion of some of the outliers

All of the 0.1% of proteins which fall outside the knot bound are of length at most 17, with one example highlighted in Fig 6A. Though this protein falls outside the theoretical knot bound, we find that its backbone is unknotted according to the KnotProt database [47]. For this protein, its few secondary structure elements are coiled with consistent chirality leading to high writhe. This kind of systematic entanglement is difficult to maintain for larger proteins, therefore there are no proteins of length greater than 17 outside the theoretical knot bound. In the next section we discuss the existence of helical substructures, and we see that though it is possible to have small sections with high writhe, this cannot be consistently maintained for greater lengths. This is consistent with the example in Fig 6D which had a helical substructure, but this systematic entanglement was not maintained for the whole structure.

## Helical super-secondary structures

Often when studying tertiary structure of proteins, we are interested in the folding of specific domains. For example, the CATH database provides information on protein domains that have a clear evolutionary link. These links are based on very specific folding motifs, with strict criteria on the number, length, and orientation of the secondary structural elements of the fold. One such example is the Rossmann fold [48], which consists of six $\beta$-strands, forming an extended $\beta$ sheet, where the first three strands are connected via an $\alpha$ helix, giving an alternating $\beta - \alpha - \beta - \alpha - \beta$ pattern. It has been noted [49] that this initial alternating $\alpha - \beta$ segment is the most conserved aspect of the Rossmann fold. It is also worth mentioning here that an alternating $\alpha - \beta$ motif was key in the globally helical proteins discussed above, *e.g.* Fig 6B. Crucially we have seen in our introductory example (Fig 1) that this secondary motif is present in the helical super-secondary structure of both the Rossmann Fold and TIM-barrel. A TIM-

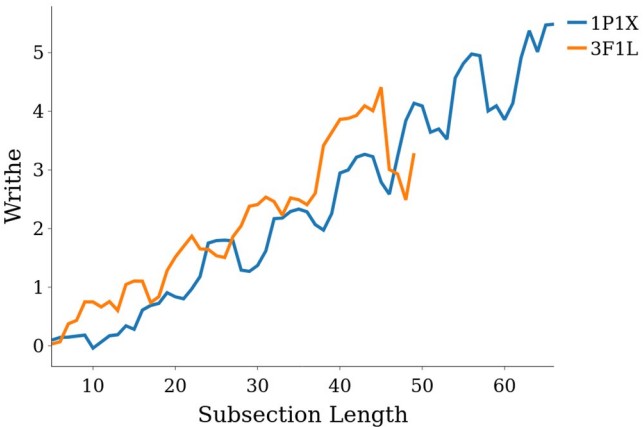

**Fig 8. A comparison of the writhe profiles of two similarly helical protein structures.** In blue, the TIM-Barrel domain 1P1X, in orange, the Rossmann fold domain 3F1L.

barrel domain consists of 8 $\alpha$-helices and 8 parallel $\beta$-strands, which alternate along the backbone. The arrangement of the $\beta$-strands on the inside of the barrel is a key aspect of the stability of this structure [50]. Plots of $Wr(\mathcal{C}_{1n})$ as a function of length for both a Rossmann Fold and TIM Barrel are shown in Fig 8. One can see that the overall gradient of growth is the same for both structures. We see between sections 5 and 42 the writhe in both cases grows to about 3.2, which gives a gradient of 0.0865 (to 3.*s.f*), not too far off the gradient of 0.12 used for the linear bounding curve in Fig 6. It is worth noting here that the examples highlighted are what are known as tandem repeat domains [46], perhaps indicating that large scale linear growth in writhe could be a quick indicator of these domains. The SWRITHE function `find_helical_sections` can identify these domains and highlight them on the SKMT smoothed curve. The routine used for this identification is included in S1 Text, here it suffices to say the code searches for significant rises in writhe for which a linear curve provides a sufficiently good fit, which implies it is (roughly) super-helical.

With this method of identifying helical domains, we can investigate how close these helical subsections are to the geometry indicative of the linear upper bound in Fig 6 (as exemplified by the super-helical structure shown in Fig 6D). We identified the helical subsections of length 20 or more present (if any) in each of the proteins in our data set, then computed the gradient of their writhe profile for this subsection. We excluded values less than 0.05 classing them as insufficiently helical, and unlikely to contribute much to the overall writhe of the full backbone. The distribution of these gradients is shown in Fig 9, with a clear bimodal distribution

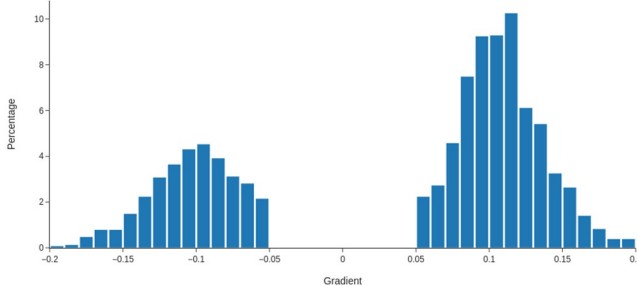

**Fig 9. The distribution of gradients for linear subsections of the writhe profiles of SKMT smooth backbones from our representative sample of the PDB.**

peaking between around the value of 0.12. There are significantly more positive gradients, and the number of gradients significantly above the 0.15 value are relatively sparse. So the helical geometries seen in the Rossman fold and TIM barrel proteins (as well as the super-helical structure shown in Fig 6D) are indicative of the dominant helical geometry found throughout the data set.

## Comparing structures using $Wr(\mathcal{C}_{ij})$

The similarity between the TIM barrel and Rossmann fold examples of the previous section opens the question as to whether we can identify other surprising tertiary structural similarities by comparing writhe profiles. To test this we create the following similarity metric for subsections $\mathcal{C}^1_{ij}$ and $\mathcal{C}^2_{lk}$ of SKMT smoothed curves $\mathcal{C}^1, \mathcal{C}^2$. We ensure $l - k = j - i$ so that the regions compared are the same size. We measure their similarity $S(\mathcal{C}^1_{ij}, \mathcal{C}^2_{lk})$ as follows

$$S(\mathcal{C}^1_{ij}, \mathcal{C}^2_{lk}) = \frac{1}{j-i-4} \sum_{m=4}^{j-i} \frac{1}{0.24m} |Wr(\mathcal{C}^1_{i,i+m}) - Wr(\mathcal{C}^2_{l,l+m})|. \tag{8}$$

This metric measures the mean absolute difference in writhe of all subsections $\mathcal{C}^1_{i,i+m}, \mathcal{C}^2_{l,l+m}$ relative to the typical linear writhe growth $0.12L$ we have seen for (SKMT smoothed) proteins. The factor features as $1/(2*0.12L)$ to account for the maximal observed difference of subsections of opposing sign writhe as a function of their length. We only consider $m \geq 4$ as the value of writhe for smaller sections is not considered meaningful. This metric is applied to all similar size $(j - i)$ subsections of the two structures, with a minimum length of 10 (imposed to focus on relatively large scale meaningful similarity). We then select the largest disjoint subsets of the two molecules which have $S(\mathcal{C}^1_{ij}, \mathcal{C}^2_{lk})$ less than some specified tolerance $s_0$. For example, a value of $s_0 = 0.05$ would indicate the average difference is less than 5% of the typical empirically observed growth in writhe difference. For context a visual depiction of what 5% and 10% similarity for some example subsections is shown in Fig 10.

An example visualisation of the results obtained applying this similarity metric is provided in Fig 11A and 11B, for the structures 1P1X (TIM Barrel) and 3F1L (Rossmann fold) with $s_0 = 0.1$. The largest similar subsections at this cutoff are $\mathcal{C}^1_{12,61}, \mathcal{C}^2_{0,49}$, covering 74% and 100% of 1P1X and 3F1L respectively, with these mutually similar sections highlighted in green.

The value $s_0$ for which sections satisfying $S(\mathcal{C}^1_{ij}, \mathcal{C}^2_{lk}) < s_0$ are classed as similar is an important one and its value will depend on what the user wants to consider as similar. In Fig 11C we chart the percentage similarity as a function of $s_0$ for 3F1L and 1P1X up to $s_0 = 0.2$. There is a sharp rise between 0.02 and 0.05, then a more steady rise. By 0.1 all of 3F1L (the smaller molecule) is considered the same as about 75% of 1P1X. As indicated in the introduction there seems to be some physical link between the CATH families to which these two protein belong so this similarity is potentially meaningful.

In the SWRITHE notebook the function `compare_molecules` can be used to compare two smoothed SKMT backbone curves with the argument for $s_0$ optional. A default value of 0.05 is suggested, and is the value used for the following results.

**Investigating the Rossmann fold -TIM barrel relationship.** To investigate how consistent this apparent relationship between Rossmann fold and TIM barrel domains is, we applied this routine to compare the Rossmann Fold domain 3F1L to all other proteins in our data set. We restrict to the cases where both structures are classed as similar for >80% of their length, so that we are looking at structures which are globally very similar to 3F1L. We find 112 such cases. A comparison to the CATH classification of these proteins shows that 62.5% of them are

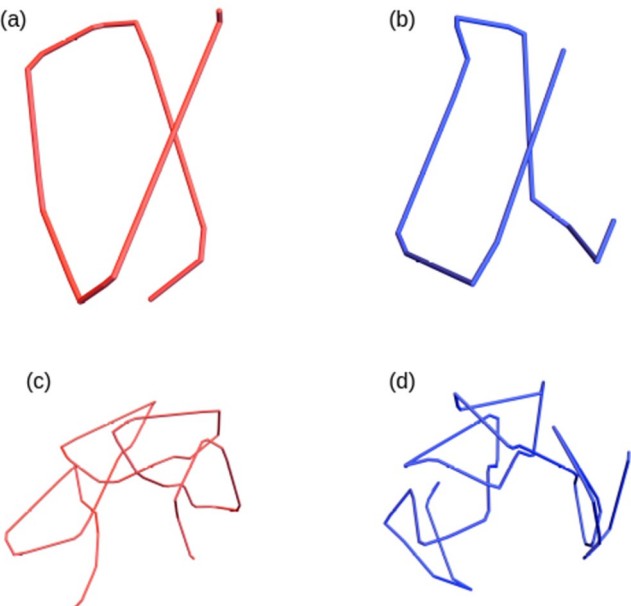

**Fig 10. Examples of curve sections sharing a 5% similarity by our comparison metric in (a) and (b) and a 10% similarity in (c) and (d).** Note the single helical loop in (a) and (b) are very uniform, whereas the four helical loops in (c) and (d) are less coherent, especially for (d).

classed as Rossmann folds (as one would expect) and 6.3% as TIM barrel domains, seemingly strengthening the structural similarity relationship between these fold types. For context 18.3% of the full database is classed as a Rossmann fold domain, and 3.4% as TIM barrel.

**Another example comparison sweep, a knotted protein.**   We also performed a similarity sweep across the database for the trefoil knotted protein 2RH3 highlighted in Fig 6C. This

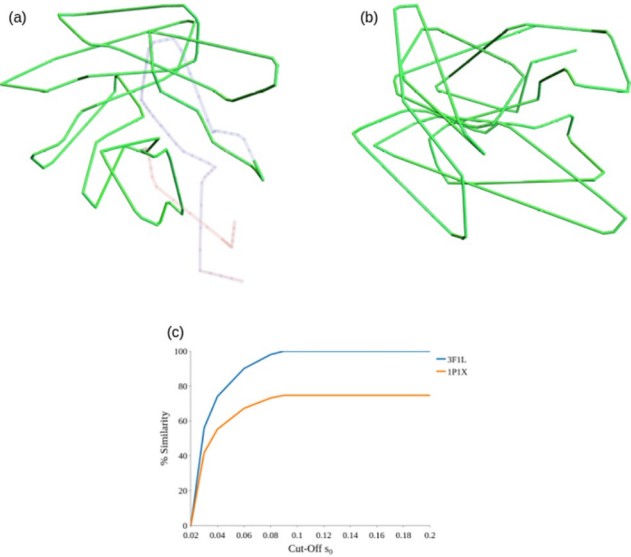

**Fig 11. Visualisations of the similarity metric $S(\mathcal{C}_{ij}^1, \mathcal{C}_{lk}^2)$ for the example Rossmann Fold (3F1L) and TIM Barrel (1P1X) domains.**

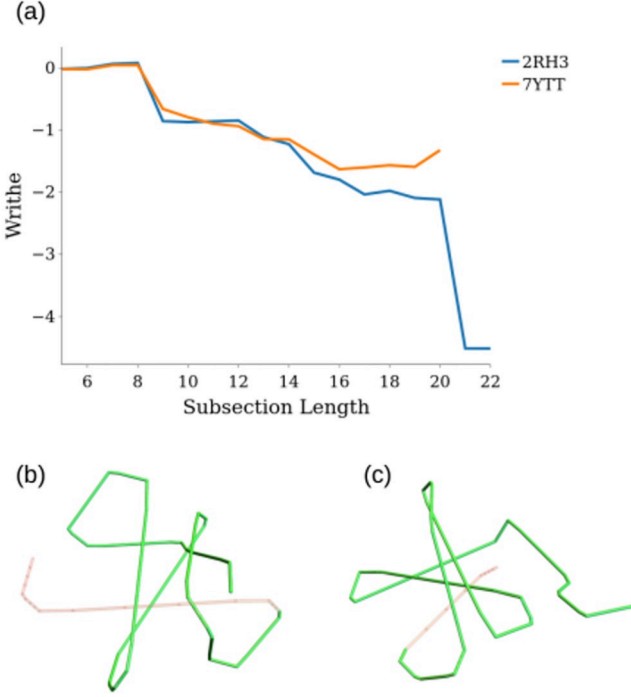

**Fig 12. The writhe profiles and mutually similar sections of the trefoil knotted 2RH3 and unknotted 7YTT.**

produced just one match above 80% similarity, PDB entry 7YTT. The matched subsections $(\mathcal{C}_{1,19})^1$, $(\mathcal{C}_{1,19})^2$ cover 82% and 90% of 2RH3 and 7YTT respectively. The writhe profiles of these two proteins is shown in Fig 12. One can see that these profiles are very similar, except for the sharp (negative) jump in writhe for 2RH3 with the inclusion of its final subsections. This sharp jump is due to the knotted nature of this backbone, with the C-terminus threading through the rest of the structure to form the trefoil. By contrast we find, using the KnotProt identification software [47], that the backbone of 7YTT is unknotted. In Fig 12B we see for 2RH3 the red C-terminus threads through the green section of curve (which is mutually similar for 2RH3 and 7YTT). In Fig 12C however, the C-terminus of the 7YTT backbone resides on the outside of the structure, which prevents it from being classed as a knotted structure. The metric has therefore identified two structures that differ only by their C-terminus threading. This similarity is missed by the standard distance based measures: jFATCAT [12] gives an RMSD of 4.26Å and TM-align [11] a TM-score of 0.23. Also the CATH classification misses this apparent similarity, in fact 7YTT has no classification. This highlights the potential in this comparative metric for identifying similar protein folds which are missed by other classification methodologies.

**A lonely Rossmann Fold.** To conclude this section on the similarity metric, we consider another example Rossmann Fold domain, PDB entry: 4QFB. Performing a sweep across our database for structures with over 80% similarity we find just one example, PDB entry 6GN5. The matched subsections are $\mathcal{C}_{21,36}^1$, $\mathcal{C}_{1,16}^2$ and $\mathcal{C}_{3,16}^1$, $\mathcal{C}_{21,34}^2$ which cover 81% and 88% of 4QFB and 6GN5 respectively. From the highlighted sections in Fig 13 we can see shared large scale helical subsections, much more uniform in the case of 4QFB. Using the Search by Sequence function on the CATH website, there are no domain matches for the FASTA sequence of 6GN5. That is because, on the secondary structure level, it does not meet the strict criteria to

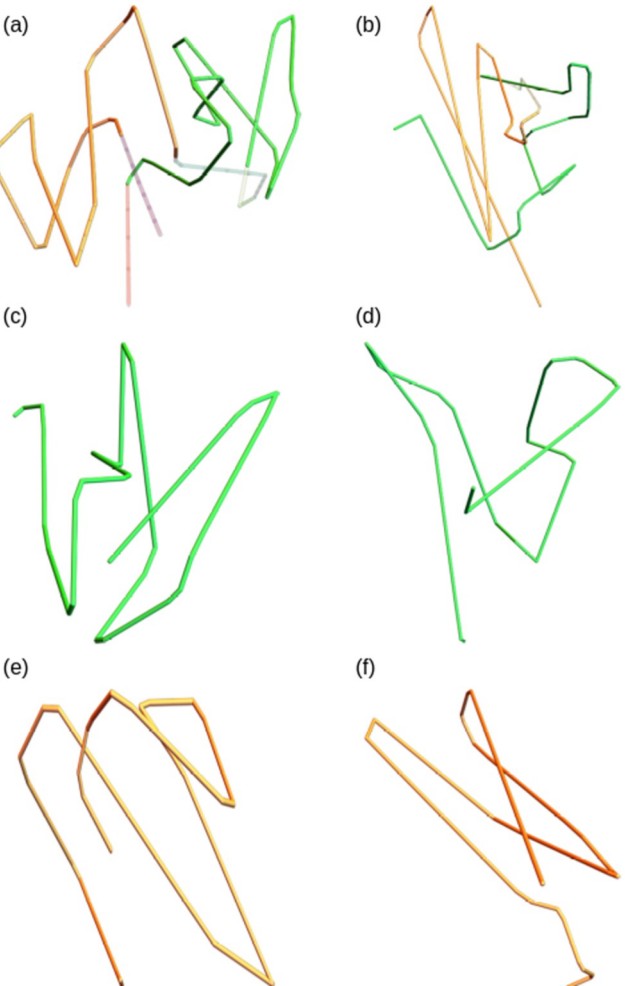

**Fig 13. The mutually similar sections of the Rossmann Fold domain 4QFB and unclassified 6GN5.**

be classified as a Rossmann fold domain. However, on the tertiary structure level, there are some clear similarities between itself and an example Rossmann fold domain. It is out of the scope of this paper to investigate this issue any further, *i.e.* whether it is a meaningful comparison, we note the expression system of both of these proteins is E-coli, so it is possible there is some functional similarity also. For now we highlight again that our comparison metric can be used to highlight potential relationships which might be missed by standard methods. Indeed, using the alignment based methods discussed in the introduction no significant similarity was found: jFATCAT gives an RMSD of 7.61Å and TM-align a TM-Score of 0.31.

**Try it for yourself.** The function `compare_molecules` in the SWRITHE notebook can be used to compare two structures from the PDB, it returns a list of the mutually similar sections and the percentage similarity in each case. The value of the cut-off $s_0$ is set at 0.05 by default but can be altered by the user. In the notebooks it is shown how the user can then apply further selection criteria, such as the requirement for the mutually similar sections to cover over 80% of both proteins, as in the previous examples. There are many other possibilities, for example searching for 100% coverage of one protein by the similar sections, but placing no restrictions on the coverage of the other. This would allow one to see if a particular structure is

similar to a subset of another, as is the case for our introductory Rossmann Fold and TIM-Barrel example.

## An empirical bound on the *acn* of proteins

We now investigate the distribution of average crossing number $acn(\mathcal{C})$ cross our dataset, and show there are apparent empirical constraints on this measure of complexity for protein structures. The distribution obtained is shown in Fig 14. There are a number of critical aspects of this plot to highlight.

1. 99.5% are bound from above by a linear growth of complexity $acn(\mathcal{C}) = L$. This is commensurate with the fact we saw in the previous section that a lot of complexity in protein structures arises from helical geometries.

2. 98.9% have an *acn* measure above the curve $(L/7.5)^{1.6} - 3$, a fit obtained by sight. This implies a possible empirically derived lower limit on the amount of complexity with respect to secondary structure.

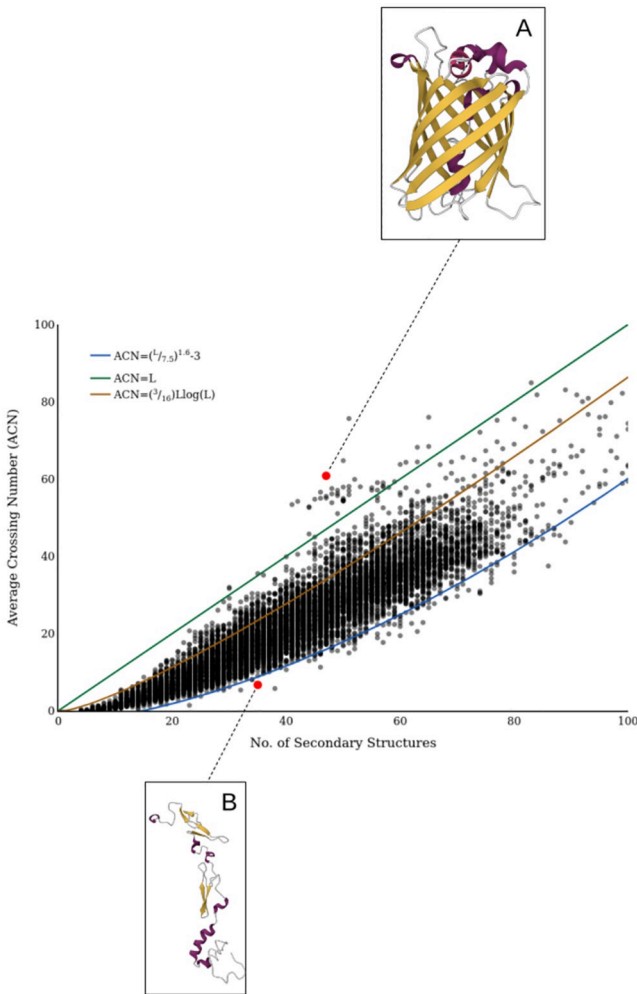

**Fig 14. The distribution of *acn* for the SKMT smoothed backbones of a representative sample of $> 10,000$ proteins.** In blue, an empirically determined lower bounding curve. In orange, the $O(L \log L)$ growth in *acn* as in [51]. In green, linear growth in *acn* with respect to length. Inset: A: PDB Entry 3EVP. B: PDB Entry 1DAN.

3. A curve $(3/16)L \log(L)$ also acts as a reasonable upper bound, with 91.4% of the data falling below this curve.

On the third point, in [51] it is shown that $acn(\mathcal{C})$ for an equilateral random walk $\mathcal{C}$ of length $n$ grows like $n\log(n)$. This bound is further studied in the context of proteins in [36] where the $acn$ of a sample of proteins is shown to follow this same length scaling. The coefficient of 3/16 determined in [36] is seen to fit closely for small proteins up to length 15. However, given our specific definition of length, and the fact that the SKMT smoothed backbone is clearly not equilateral, we can expect quite a different distribution of $acn$ in our context. In particular, the long rigid secondary structures could act as a barrier to entanglement, unless arranged in such a systematic way as to maximise entanglement, as is the case for many helical structures. For example the signaling protein seen in Fig 14A (a representative of the cluster of proteins whose $acn$ exceeds the $O(L \log L)$ growth) forms a large $\beta$-barrel structure. It is clear that a uniform random walk would be highly unlikely to achieve this specific helical and symmetric structure.

**Falling below the empirical complexity.**   In order to use the $acn$ in a predictive capacity, it is important to understand how strict this empirical lower bound is. In Fig 14B we highlight an illustrative example of the subset of proteins whose $acn$ falls below the empirical bound displayed in blue. It is a single chain from a complex of active site inhibited human blood coagulation factor VIIA with human recombinant soluble tissue factor. This protein forms a hetero 4-mer in its native state, with some complex entangled subunits. However, in this study we are looking to quantify the self entanglement of monomer units, so for any multimeric protein that was sampled from the PDB we extracted just the first chain from its PDB file for analysis. With 63.6% of the proteins that fall below the empirical lower bound being multimers, we are safe to use it in a predictive capacity for monomer structural predictions. The remaining 36.4% of proteins falling below the bound are due to poor secondary structure assignment. This difficulty when working with experimentally determined structures is addressed via a secondary structure assignment cleaning step discussed in more detail in S5 Fig. This simple cleaning step removes any obvious singleton $\alpha$-helix or $\beta$-strand residues, and is applied automatically in the `skmt` routine. Though there does remain some edge cases missed by this initial cleaning step, they represent 0.4% of the sample so we do not feel they render this predicted lower bound for monomer protein entanglement meaningless.

## Using the *acn* to improve BioSAXS predictions

We now highlight a concrete use of this empirical *acn* lower bound for structural prediction. We consider the gene regulatory protein SMARCAL1. This protein regulates gene transcription through the alteration of the chromatin structure around those genes [52]. There is a predicted structure from AlphaFold for this protein, seen in Fig 15A, however it has regions of low confidence, and most importantly is a poor fit to the small angle x-ray scattering data for this protein (obtained on the B21 Beamline at the Diamond light source). This is illustrated in Fig 15B where the SAXS scattering model is obtained using the method described in [53] (similar quality fits were obtained using the FOXS web server [54] as a check). For those readers not familiar with the BioSAXS data analysis, the factor $q$ on the x-axis measures the momentum transfer and the vertical axis shows the logarithm of the observed scattering intensity $I(q)$. The lower $q$ range corresponds to larger scale structural information and the resolution increases with higher $q$. Fitting low $q$ range is therefore of paramount importance as a small discrepancy with the data there can mean the overall shape of the molecule is wrong. By contrast at higher $q$ discrepancies in a fixed prediction are less meaningful. Thus, for this illustrative example we stick to the $q \in [0, 0.15]$ range. A rough rule of thumb for these experiments is

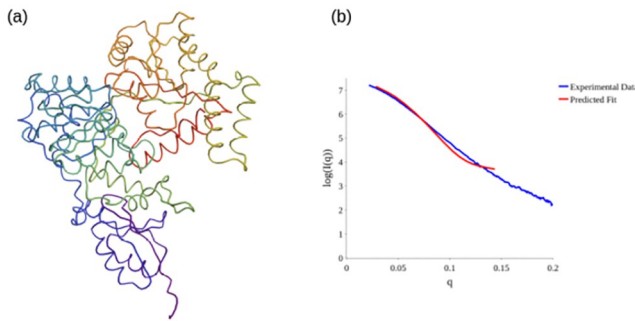

**Fig 15. The AlphaFold predicted structure for Human SMARCAL1 has regions of low confidence, and the fit to the scattering data suggests it opens out in solution.**

that globular conformations have a hill shape low $q$ SAXS curve whilst pill-like structures have a flatter low $q$ (see *e.g.* Fig 3 of [55]). Fig 15 indicates the AlphaFold prediction is too globular and the structure likely opens out somewhat in solution.

Using the constrained backbone algorithm [53], a new potential structure was produced to fit the BioSAXS data. By systematically varying the geometry of individual secondary structures of the $C^\alpha$ backbone according to Ramachandran like constraints, a locally plausible structure is predicted. However, the original method had no constraint on the plausibility of the overall tertiary structure. This is a potential issue as the inverse scattering problem is not well posed [56] and multiple differing predictions for the data can be made. Though a prediction could in theory be tested using an all atomistic physics model, this is an extremely time consuming and intensive process. By contrast the *acn* calculations are of lower complexity than the scattering calculation itself, and can at the very least be used to rule out good fits to the scattering data which are unrealistically unfolded. We ran a series of fits to the SAXS data on $q \in [0, 0.15]$ and calculated the *acn* of the final prediction.

An example fit is shown in Fig 16A to "open out" quite significantly. Visually we can see the three key domains of the structure are too far apart. Since SMARCAL1 has 84 secondary structure elements, we would expect its *acn* to be at least 56.8 (to 3 *s.f*) and likely a bit above this. The original structure's *acn* is 69.5 (to 3 *s.f*). This opened out structure however has an *acn* value of 48.3 (to 3 *s.f*). We then modified the constrained-backbone algorithm of [53] with a penalty for structures whose *acn* falls below the blue bounding curve (a simple step function for this testing exercise), to produce the predicted structure seen in Fig 17A. This structure is much more globular (although less so than the original structure as expected from the shape of the scattering data), and its *acn* is 59.2 (to 3 *s.f*) which is above the bound. It should be clear

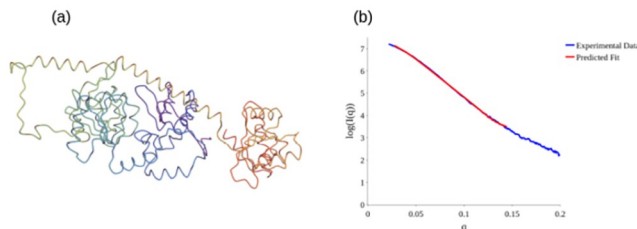

**Fig 16. Examples of potential backbone structures which fit the SMARCAL1 data very well but are unrealistically unfolded according to the empirical bound on *acn*.**

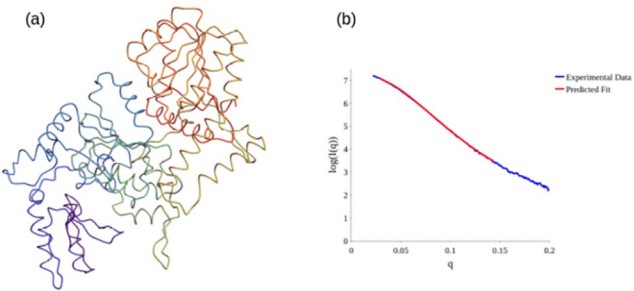

**Fig 17. A predicted structure for SMARCAL1 which is both a good fit to the scattering data and is realistically folded according to the empirical *acn* bound.**

this is not sufficient to make a clear prediction that the outcome is a plausible structure for the protein, further tests such as MD simulations and other experimental and statistical analysis would be needed for this. This is merely an example test of the potential efficacy of the *acn* measure as a means of providing a computationally efficient additional constraint on the tertiary fold search space.

## Discussion/conclusion

In this study, we detail a series of metrics and methods for utilising the writhe and average crossing number as a measure of tertiary structure entanglement for protein backbone curves. These metrics are applied to a smoothed representation of the $C^\alpha$ backbone, which treats $\alpha$-helices and $\beta$-strands as rigid units and simplifies linker sections whilst preserving the fundamental topology of the structure. This method of smoothing we call the SKMT (Secondary KMT) algorithm, an adaptation of the KMT algorithm which is a known method for reducing a curve to a minimal representation which preserves its tertiary entanglement. We argue that this method of smoothing best captures the link between potential complexity of entanglement and secondary structure of a protein.

In this study, we have shown that the writhe of these smoothed protein backbones is bounded with respect to the number of secondary structure elements. This is done by applying the metric to a set of >10000 sequentially unique (at the 70% level) high quality structures. In particular, a linear bounding curve of growth in the writhe with respect to the number of secondary structures, with a gradient of 0.12, contains 97.8% of protein structures. We show that this linear scaling is directly linked to the geometry of globally helical protein structures. This same helical geometry is consistently found in subsections of other proteins. This helical geometry is a method of systematically building up writhe in such a way as to produce thermally stable structures with strong inter sheet bonds (TIM-barrel structures being a common example). Conversely, many proteins exhibit zero net entanglement whilst exhibiting significant complexity, across all length scales.

We then study the distribution of the average crossing number for proteins, again establishing empirical bounds on this measure of complexity as a function of the number of secondary sections of the protein. We show again that linear growth is a good upper bound on complexity, strengthening the case that complexity of entanglement in medium to large proteins is due to large scale tertiary helical geometries. We also provide an empirical lower bound on the *acn* of proteins, which we show is effective when used as a constraint for realistic and efficient searches of potential structure landscapes. A concrete example of how this can be applied to structural predictions from BioSAXS data is detailed.

Alongside this work we provide an ipython notebook where one can

1. Apply the SKMT secondary backbone smoothing algorithm to a specified PDB structure, using the `skmt` function.

2. Compute the writhe and average crossing number of the smoothed backbone of a given protein using the `calculate_writheFP` function.

3. Highlight helical subsections of a given structure with the `view_molecule_helical` function.

4. Identify subsections of two proteins of similar geometry through a comparison metric which calculates the total percentage similarity of the writhe distribution of two protein backbones (SKMT smoothed). We showed it can identify similar tertiary structures missed by other classification methods (and indeed sometimes unclassified in the CATH database). A protein can be tested against our database in only a few minutes using the `compareToDatabase` function in the SWRITHE package.

## Supporting information

**S1 Fig. The potential for missing N-termini.** It is often the case that the N terminus may be missing from a PDB entry, or a purification tag is used to experimentally determine the structure. To study the potential effect of these foibles on our writhe calculations, we randomly cut between $10 - 20$ residues from the start of each PDB file before applying the SKMT smoothing. The writhe of the SKMT backbones are calculated as before and their values are plotted against the length (number of points) of the smoothed curve. In S1 Fig. The distribution is overlaid with the standard distribution of writhe presented in Fig 6 in red for comparison. The overall shape of the two distributions is the same, with the majority of points falling well within the linear bound. Indeed, with the cut residues we find that 98.3% of the data falls within the linear bound, compared with 97.9% in the original case. Since the cutting of 10–20 residues from the N terminus end of each protein means removing 1 or 2 points from the SKMT curve, there is little change to the nature of the overall entanglement.
(TIF)

**S2 Fig. The distribution of writhe against arclength.** In S2 Fig we plot the distribution of writhe of the SKMT smoothed backbones against their arclength. Since this definition of length is proportional to our choice of length, the empirically determined linear bounding curve performs similarly well for this distribution containing 98.1% of the data. However, it appears there is more of a gap between the bounding line and the main distribution for SKMT curves of arclength between 150–250, compared to the equivalent section of the plot in the main text (Fig 6). In some sense the bound appears slightly less "tight". As a heuristic measure of the tightness of the linear bound, we compute the distance from the linear bound for the closest value at each length, then take the average of these distances. In the case of the arclength distribution, this average closest distance is 3.1 whereas for the SKMT length distribution it is just 1.2, indicating that our choice of length better captures the length scaling relationship of entanglement we are aiming for in this study.
(TIF)

**S3 Fig. The distribution of writhe against number of amino acids.** In S3 Fig we plot the distribution of writhe of the SKMT smoothed backbones against the number of amino acids of the respective protein. An empirically determined linear bounding curve fit to contain the same proportion of the data as in Fig 6 is shown in green. The average closest distance to the

bound for this fit is 2.9, indicating that the tightness of this fit is not optimal.
(TIF)

**S4 Fig. The distribution of writhe for uniformly sampled backbones.** In S4 Fig we see that there is no clear linear relationship between writhe and length for backbones that are uniformly smoothed by sampling every $n = 4^{th}$ amino acid (a similar lack of clear relationship was obtained for $n = 3, 5, 6, 7$). In the SKMT case, a linear gradient of 0.12 is sufficient to capture 97.9% of the data, with a uniform spread of outliers across all lengths. Here, a linear gradient of 0.12 achieves a similarly good fit containing 98.8% of the data, however it is an overestimation for positively entangled small proteins, and negatively entangled large proteins, with an average closest distance in these regions of 4.6. In green, we see a linear bounding curve with gradient of 0.08. Though the average closest distance for this fit is good at 0.19, only 89.1% of the data lies within this bound. The SKMT smoothed approach therefore performs better at representing the relationship between potential complexity and secondary structure.
(TIF)

**S5 Fig. Secondary Structure Cleaning.** During the initial study some proteins had *acn* well below the empirically determined lower bound despite appearing maximally entangled. This was due to the fact that our definition length in this study is dependent on the number of SSEs. There were some anomalies in the secondary structure classification of these structures, in particular they contained many single amino acid long SSEs. As a result, the length assigned to these proteins was much greater than the realistic number of SSEs. A simple routine removing any of these single amino acid $\alpha$-helices or $\beta$-strands from between linker sections allows these proteins to be located well above the blue minimum bound curve. This cleaning routine was applied to the full sample from the PDB before computing the distribution of *acn* against the SKMT length. Its effect on the percentage of structures falling below the lower bounding curve was significant as can be seen in S5 Fig. This initial pitfall acts as a reminder of the need to be careful when working with PDB files and secondary structure assignment. Though our initial use for the lower bound on entanglement is for identifying unrealistically folded structural predictions (as highlighted in the main text), it could also serve a purpose in spotting poor secondary structure assignment. This secondary structure cleaning is now performed as standard in the SKMT algorithm, via the `simple_ss_clean` function.
(TIF)

**S1 Text. Routine for identifying helical subsections.** The routine used to identify linear subsections of the writhe profile is as follows.

1. First perform a LOWESS (locally weighted scatterplot smoothing) on the writhe data.

2. Then, for all subsections of length greater than 20, compute the gradient of the writhe profile of this subsection.

3. If this gradient is larger than 0.05 (i.e within 50% of the maximally observed linear growth), this subsection is potentially helical.

4. For a potentially helical subsection $\mathcal{C}_{i,j}$, we then check that there is no change in the sign of the gradient $Wr(\mathcal{C}_{i,i+k})/Wr(\mathcal{C}_{i,i+j})$ for all $k = 1, j - i$. This ensures we identify subsections with consistent linear growth in writhe.

5. Finally, we output the largest disjoint subsections satisfying the above criteria.
(PDF)

## Author Contributions

**Conceptualization:** Robert Rambo, Christopher Prior.

**Data curation:** Arron Bale, Robert Rambo.

**Formal analysis:** Arron Bale, Christopher Prior.

**Investigation:** Arron Bale.

**Methodology:** Arron Bale, Christopher Prior.

**Resources:** Robert Rambo.

**Software:** Arron Bale, Christopher Prior.

**Supervision:** Robert Rambo, Christopher Prior.

**Validation:** Christopher Prior.

**Writing – original draft:** Arron Bale, Christopher Prior.

**Writing – review & editing:** Arron Bale, Christopher Prior.

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
