## [Decision Letter · Decision Letter 0]

29 Jul 2023

Dear Mr Bale,

Thank you very much for submitting your manuscript "Novel topological methods for identifying surprising protein tertiary structure relationships." for consideration at PLOS Computational Biology.

As with all papers reviewed by the journal, your manuscript was reviewed by members of the editorial board and by several independent reviewers. In light of the reviews (below this email), we would like to invite the resubmission of a significantly-revised version that takes into account the reviewers' comments.

Specifically, please follow the reviewer's suggestions to significantly shorten the manuscript. In addition, please provide the link to the github repository.

We cannot make any decision about publication until we have seen the revised manuscript and your response to the reviewers' comments. Your revised manuscript is also likely to be sent to reviewers for further evaluation.

Sincerely,

Dina Schneidman

Academic Editor

PLOS Computational Biology

Nir Ben-Tal

Section Editor

PLOS Computational Biology

Reviewer's Responses to Questions

**Comments to the Authors:**

Reviewer #1: Bale and co-authors present a "novel" method to measure entangled structures in proteins. Overall, I think the paper is very long and very confusing. It's unclear what is the main goal of the authors. I don't think 29 figures are necessary to prove the authors' points. I suggest major rewriting of the paper before publication.

I genuinely think that published in this form, it will put off a lot of readers.

I suggest the authors on focusing on one/two points and narrow the story around those.

Use the SI for supporting information that are not key to the story.

Finally, the code is not shared with the referees. It would be important that the authors put on paper the website of the repository.

1. I agree with the authors about the "somewhat lengthy introduction". I think it's nice to give an overview of the topic, but I also think this introduction distracts from the main objectives of the paper. I suggest the authors to try to condense to the key issues.

The style of the intro and of most of the paper gives the reader the feeling that this paper has been put together without much thought, and almost as a descriptive "shopping list" of "stuff" that the authors think are relevant to this topic.

The title is not very informative

"Novel topological methods for identifying surprising protein tertiary structure relationships."

can mean anything. I suggest the authors to convey a bit more precisely their discovery.

2. Fig. 9b should have labels on the axes and the plots with the scattering data have no units.

3. Many plots have different style of presentation, font, etc. I'd say that this gives the impression that the paper has been put together in a rush and somewhat unprofessionally.

4. This paper has 29 figures - I am sure that some of them are not necessary and can be put in SI to streamline the discussion. For a reader, especially someone who wants to use their code, it's really hard to understand what is important and what is not.

Spelling/typos

- "visualise a proteins global structure"

- please add "fig." where needed e.g. "of a ribbon diagram as seen in (Fig)1."

- missing reference in sentence "We took the largest monomer unit from each PDB and obtain a simplified backbone using the SKMT algorithm described in ."

- typo in "We too the “length” L"

- other typos or poor grammar are scattered around the text, again signature of a rushed paper.

For instance, "This we show how it can .. structures."

Reviewer #2: Uploaded as a pdf

Reviewer #3:

My concern is the lengthy report form of the manuscript and its relatively week conclusions.

Remarks:

1) Both smoothing of the backbone prior to calculation of Gauss Integrals as well as a study of their upper bound as function of chain length is not new. It is a major part of Peter Røgen: Evaluating protein structure descriptors and tuning Gauss integral based descriptors 2005 J. Phys.: Condens. Matter 17 S1523

Developed precisely (as in the manuscript at hand) to establish relationship at tertiary level (for transition-state ensemble structures) in Protein folding and the organization of the protein topology universe, Kresten Lindorff-Larsen et all. TIBS 30,1 2005

2) Re Figure 9a: Why does writhe jump by +-4 at self-intersections? I believe it should be by +-2

Michael Livitt used this to detect self-intersections in Protein Folding by Restrained Energy Minimization and Molecular Dynamics, J Mol. Biol. (1983) 170, 723-764

3) The cutting of protein chains into domains (CATH) may vary and for full chains sometimes the N-terminal is missing in pdb-files. The authors therefor need to take into consideration what the deletion of the first 10-20 residues does to all the calculations of writhe (Figures 5, 6 7 10,...) from the beginning to a varying endpoint. What happens if another starting point is chosen?

**Have the authors made all data and (if applicable) computational code underlying the findings in their manuscript fully available?**

Reviewer #1: **No: **no link to repo

Reviewer #2: Yes

PLOS authors have the option to publish the peer review history of their article (what does this mean?). If published, this will include your full peer review and any attached files.

Reviewer #1: No

Reviewer #2: No
---

## [Decision Letter · Decision Letter 1]

6 Nov 2023

Dear Mr Bale,

We are pleased to inform you that your manuscript 'The SKMT Algorithm: A method for assessing and comparing underlying protein entanglement' has been provisionally accepted for publication in PLOS Computational Biology.

Best regards,

Dina Schneidman

Academic Editor

PLOS Computational Biology

Nir Ben-Tal

Section Editor

PLOS Computational Biology

Reviewer's Responses to Questions

**Comments to the Authors:**

Reviewer #1: The authors have addressed all my comments and I think the paper readability has improved a lot.

As a side note, I flag this recent paper (to appear in Soft Matter) to the authors. https://arxiv.org/abs/2305.11722

We have used a "local writhe" feature as input to a neural network that can identify different knots.

I think it could potentially be useful for protein structure and entanglement recognition.

Specifically, eq(1) defines a generalisation of the writhe into a "segment-to-segment" map which, if computed on SKMT-processed structures, could identify interesting entanglement features.

I am mentioning this in case the authors are interested in using our local writhe feature for proteins.

Reviewer #2: The authors address all of the comments and substantially improved the paper.

**Have the authors made all data and (if applicable) computational code underlying the findings in their manuscript fully available?**

Reviewer #1: Yes

Reviewer #2: Yes

PLOS authors have the option to publish the peer review history of their article (what does this mean?). If published, this will include your full peer review and any attached files.

Reviewer #1: **Yes: **Davide Michieletto

Reviewer #2: No

---

## [Editor Report · Acceptance letter]

17 Nov 2023

PCOMPBIOL-D-23-00899R1 

The SKMT Algorithm: A method for assessing and comparing underlying protein entanglement

Dear Dr Bale,

I am pleased to inform you that your manuscript has been formally accepted for publication in PLOS Computational Biology. Your manuscript is now with our production department and you will be notified of the publication date in due course.

With kind regards,

Anita Estes
